# Characteristics, Source Contributions, and Source-Specific Health Risks of PM_2.5_-Bound Polycyclic Aromatic Hydrocarbons for Senior Citizens during the Heating Season in Tianjin, China

**DOI:** 10.3390/ijerph19084440

**Published:** 2022-04-07

**Authors:** Nan Zhang, Chunmei Geng, Jia Xu, Liwen Zhang, Penghui Li, Jinbao Han, Shuang Gao, Xinhua Wang, Wen Yang, Zhipeng Bai, Wenge Zhang, Bin Han

**Affiliations:** 1State Key Laboratory of Environmental Criteria and Risk Assessment, Chinese Research Academy of Environmental Sciences, Beijing 100012, China; zhangnan01@craes.org.cn (N.Z.); gengcm@craes.org.cn (C.G.); carol3233@126.com (J.X.); wangxh@craes.org.cn (X.W.); yangwen@craes.org.cn (W.Y.); baizp@craes.org.cn (Z.B.); 2Department of Occupational and Environmental Health, School of Public Health, Tianjin Medical University, Tianjin 300070, China; zhangliwen@tmu.edu.cn; 3School of Environmental Science and Safety Engineering, Tianjin University of Technology, Tianjin 300384, China; lipenghui406@163.com; 4School of Quality and Technical Supervision, Hebei University, Baoding 071002, China; jinbaobaohan@163.com; 5School of Geographic and Environmental Sciences, Tianjin Normal University, Tianjin 300387, China; shuang1gao@163.com; 6Particle Laboratory, Center for Environmental Metrology, National Institute of Metrology, Beijing 100022, China

**Keywords:** PM_2.5_, polycyclic aromatic hydrocarbons, personal exposure, positive matrix factorization, health risk

## Abstract

Polycyclic aromatic hydrocarbons (PAHs) have carcinogenic impacts on human health. However, limited studies are available on the characteristics, sources, and source-specific health risks of PM_2.5_-bound PAHs based on personal exposure data, and comparisons of the contributions of indoor and outdoor sources are also lacking. We recruited 101 senior citizens in the winter of 2011 for personal PM_2.5_ sample collection. Fourteen PAHs were analyzed, potential sources were apportioned using positive matrix factorization (PMF), and inhalational carcinogenic risks of each source were estimated. Six emission sources were identified, including coal combustion, gasoline emission, diesel emission, biomass burning, cooking, and environmental tobacco smoking (ETS). The contribution to carcinogenic risk of each source occurred in the following sequence: biomass burning > diesel emission > gasoline emission > ETS > coal combustion > cooking. Moreover, the contributions of biomass burning, diesel emission, ETS, and indoor sources (sum of cooking and ETS) to PAH-induced carcinogenic risk were higher than those to the PAH mass concentration, suggesting severe carcinogenic risk per unit contribution. This study revealed the contribution of indoor and outdoor sources to mass concentration and carcinogenic risk of PM_2.5_-bound PAHs, which could act as a guide to mitigate the exposure level and risk of PM_2.5_-bound PAHs.

## 1. Introduction

Polycyclic aromatic hydrocarbons (PAHs) are persistent organic pollutants with multiple fused aromatic rings [1,2], which are primarily emitted during incomplete anthropogenic combustion [2,3,4]. In addition, PAHs have a seasonal variation with peak values appearing in winter [3,5,6,7,8,9]. There have been sustained concerns about PAHs due to their mutagenicity, teratogenicity, and carcinogenicity [3,10,11]. Several PAHs had been declared as carcinogenic, probably carcinogenic, or possibly carcinogenic to the public [12]. Carcinogenic PAHs mainly exist in the particulate phase, especially in fine particulate matter (particulate matter with an aerodynamic diameter of less than 2.5 μm, PM_2.5_) [9,13,14,15,16,17]. Inhalational exposure to PM_2.5_-bound PAHs is inevitable because they are ubiquitous in the atmosphere. Once inhaled, PAHs induce oxidative stress and inflammation [18,19], leading to lung function decline and pulmonary impairment [20], and increase the risk of lung cancer [1,2,21]. In China, lung cancer was the leading cause of death among patients with malignant tumors, and approximately 1.6% of lung cancer cases were related to PAH exposure via inhalation [22].

The levels of personal exposure to PM_2.5_-bound PAHs are influenced by outdoor sources (e.g., biomass burning, coal combustion, and traffic exhaust) and indoor sources (e.g., cooking fumes and environmental tobacco smoking (ETS)) [3,4,23,24,25,26]. Numerous studies investigated the outdoor sources of PM_2.5_-bound PAHs [6,16,22,27,28,29], and several studies have used outdoor data as surrogates to assess PAH exposure risk [6,7,22,30], while few studies used the personal exposure data [10,14,20]. Furthermore, indoor emission of PM_2.5_-bound PAHs deserves increased attention. People spend the majority (80–90%) of their time indoors [23,31,32], and the indoor levels of PM_2.5_-bound PAHs are higher than the outdoor levels [3,4]. Moreover, compared to outdoor sources, people are exposed to indoor sources directly with less diffusion [20,33,34]. Additionally, indoor exposure to PM_2.5_-bound PAHs might considerably contribute to the PAHs burden inside the body [18]. Hence, as a link between indoor and outdoor exposure, observation of personal exposure can simultaneously consider exposure levels in different microenvironments and better characterize exposure risks.

Exposure to PAHs originated from different sources would induce diverse health risks, due to the disparate source profiles of PAHs emission and distinct toxicities of individual PAHs [3,11,16,26]. Lai et al. [24] suggested that reducing personal PM_2.5_-bound PAH exposure should address both indoor and outdoor sources, which could directly mitigate lung cancer risk [35]. Therefore, based on the source apportionment of personal exposure, the source-specific health risk is a better index to inform policy and regulatory strategies with minimum economic cost and maximum health benefits [6,36]. In addition, for the public, it is convenient to adopt intervention methods, such as using a range hood [37] in the kitchen or preventing indoor smoking to reduce indoor-generated PAHs. Although it is hard to answer the question of whether exposure to one PM source is associated with worse health effects than an equivalent exposure to other sources [36], the ratio of contribution to carcinogenic risk versus contribution to mass concentration could provide a preliminary clue. However, information on source-specific health risk from personal PM_2.5_-PAH exposure was limited [10,14], especially those from indoor PAH sources.

In this study, we hypothesize that indoor and outdoor exposure to PM_2.5_-bound PAHs might induce severe health risk for senior citizens. We conducted a sampling campaign by collecting personal PM_2.5_ samples and analyzed the PAH species associated with PM_2.5_. We characterized personal PAH exposure for the senior citizens in winter, identified the contributing sources, and estimated the source-specific risk via inhalation. In addition, we compared the source-specific mass contribution and source-attributed health risk between indoor and outdoor PAHs sources.

## 2. Materials and Methods

### 2.1. Sampling Description

#### 2.1.1. Sampling Area

Tianjin is a metropolis located in the North China Plain. According to the National Bureau of Statistics of China (http://www.stats.gov.cn/english/ (accessed on 2 August 2021)) in 2011, the population of Tianjin was 13.55 million, and 9.77% were senior citizens (>65 years old). The number of private cars was 1.55 million, and coal consumption accounted for over 50% of fossil fuel consumption. During the last decade, Tianjin has suffered from heavy air pollution due to rapid industrialization and urbanization.

In this study, we selected several residential communities located downtown near arterial roads (within 400 m) and a gas station (about 1 km), with a university and a hospital nearby. The senior citizens living in the communities were recruited as volunteers for personal PM_2.5_ sample collection.

#### 2.1.2. Participant Recruitment

During volunteer recruitment, questionnaires were distributed to the senior citizens, containing information such as name, sex, age, living habits, and health condition. Participants with a poor health condition or a social activity radius of more than 5 km were excluded, and 101 healthy senior citizens took part in sample collection. General information on the participants is listed in Appendix A.

#### 2.1.3. Sample Collection

Sample collection was performed in the winter (30 November to 12 December) of 2011. Each participant was asked to wear a backpack for a consecutive period of 24 h (08:00 to 08:00, next day). The backpack held a pump (A.P. Buck Inc., Orlando, FL, USA) inside, which was linked to a personal exposure monitor sampler (PEM-PM_2.5_, BGI Inc., Waltham, MA, USA) with a pipe; the sampler contained a 37 mm quartz filter (Pall-Gelman, Ann Arbor, MI, USA) and was placed on the shoulder strap of the backpack to collect PM_2.5_ samples near the breathing zone. The pump was calibrated using a flowmeter (Buck Inc., Waltham, MA, USA) at a flow of 4 ± 0.2 L/min. Before the sampling, the senior citizens were asked to behave as usual and carry the backpack throughout the sampling period (except while bathing and sleeping), in order to represent actual personal PM exposure levels precisely. Finally, 87 valid samples were collected for PAHs analysis, and the data were pooled together to explore exposure characteristics, source contributions, and health risks for the senior population.

### 2.2. Mass and PAH Analysis

#### 2.2.1. Mass Analysis

Before use, all quartz-fiber filters were pre-heated at 800 °C for 2 h. The filters were equilibrated in an enclosed chamber (temperature: 22 ± 1 °C; relative humidity: 35 ± 1%) for more than 48 h, exposed to a radioactive object to reduce static, and weighted using a micro-balance (sensitivity: ±1 μg) (Mettler MX5, Mettler-Toledo, Greifensee, Switzerland). The average value of three consecutive measurements was recorded for each filter.

#### 2.2.2. Polycyclic Aromatic Hydrocarbons Analysis

The method of gas chromatography coupled with mass spectrometry (GC/MS, trace 2000 GC-MS, Thermo Finnigan, Waltham, MA, USA) was applied to detect PAHs, including acenaphthene (Ace), fluorene (Flu), phenanthrene (Phe), fluoranthene (Fl), pyrene (Pyr), benz[a]anthracene (BaA), chrysene (Chr), benzo[b]fluoranthene (BbF), benzo[k]fluoranthene (BkF), benz[e]pyrene (BeP), benzo[a]pyrene (BaP), dibenz[a,h]anthracene (DahA), benzo[ghi]perylene (BghiP), and indeno[1,2,3-cd]pyrene (IND). A detailed description of analysis procedures and quality assurance/quality control is provided in Appendix A and a previous publication [38]. Briefly, the quartz-fiber filter was completely immersed into dichloromethane and extracted using an ultrasonicator. The extract was condensed to 5 mL using a rotary evaporator at first, then concentrated to 1 mL by blowing pure nitrogen, and injected into GC-MS to separate and identify the targeted PAHs.

Based on the number of carbon rings, PAHs could be classified as low molecular weight (LMW) with two or three benzene rings, middle molecular weight (MMW) with four benzene rings, and high molecular weight (HMW) with five or more benzene rings [4,27].

### 2.3. Data Analysis

#### 2.3.1. Source Apportionment

Positive matrix factorization (PMF) was developed by Paatero and has been widely used to recognize potential sources without source profiles. EPA PMF version 5.0 was released by the United States Environmental Protection Agency (US EPA, Washington, DC, USA) and could be downloaded at the website of US EPA (https://www.epa.gov/sites/default/files/2015-03/epa_pmf_5.0_setup.exe (accessed on 5 April 2022)), which was operated in this study. PMF decomposed the concentration of the targeted pollutant into a factor contribution and profile, with the following equation:(1)$$x_{ij}=\sum_{k=1}^{p}g_{ik}f_{kj}+e_{ij}$$
where *x_ij_* is the concentration of the *j*th species of the *i*th sample; *p* is the number of factors; *g_ik_* is the factor contribution (the contribution of the *k*th source to the *i*th sample); *f_kj_* is the factor profile (the concentration of the *j*th species in the *k*th source); *e_ij_* is the residual.

To determine the number of factors, non-negative factor contribution, and factor profile, PMF was adjusted to the value of *g_ik_* and *f_kj_* to minimize the *Q* value with the following function:(2)$$Q=\sum_{i=1}^{n}\sum_{j=1}^{m}{\left[\frac{x_{ij}-\sum_{k=1}^{p}g_{ik}f_{kj}}{u_{ij}}\right]}^{2}$$
where *u_ij_* is the uncertainty of the *j*th species of the *i*th sample, calculated by the following equation:(3)$$Uncertainty=\sqrt{{\left(Error\;Fraction\times concentration\right)}^{2}+{\left(MDL\right)}^{2}}$$
where *MDL* is the method detection limit of every chemical composition; for the species with a concentration less than *MDL*, the uncertainty was 5/6 of the *MDL*, and for missing data, the uncertainty was three times higher than the average concentration of the species.

The ratio of *Q*/*Q_exp_* was used to determine the number of factors. With the increase in the factor number (tested from 3 to 8), a significant decrease in *Q*/*Q_exp_* indicated an optimized model explanation with the selected factor number. In contrast, a small decrease in *Q*/*Q_exp_* suggested little model improvement with the extra factors [10]. Thus, the number of factors that should be selected after *Q*/*Q_exp_* appeared to decrease. *Q_exp_* was calculated with the following formula:(4)$$Q_{exp}=N_{sample}\times M_{good}+\frac{N_{sample}\times M_{weak}}{3}-\left(N_{sample}\times P_{factor}\right)$$
where *N_sample_* is the number of the collected samples run; *M_good_* is the number of species labeled as good; *M_weak_* is the number of species labeled as weak, and *P_factor_* is the number of factors.

The base run of PMF could start the gradient algorithm with a random starting point (random seed model) or a fixed starting point. The user guide of PMF stated that it could test whether the solution found was optimal by using many random seeds and examining whether the *Q* values are stable.

#### 2.3.2. Health Risk Assessment

The excess cancer risk via inhalation was calculated based on exposure concentration, as follows [39]:(5)Risk=EC×IUR
where *EC* is the exposure concentration (μg/m^3^); *IUR* is the inhalation unit risk (μg/m^3^)^−1^.

As the toxicity level of each PAH was different, the “*EC*” in the above equation referred to the equivalent concentration of BaP (*BaP_eq_*), which could quantify the potential toxicity of PAH exposure [40], calculated using the equation below:(6)$$BaP_{eq}=\sum_{i=1}^{n}TEF_{i}\times C_{i}$$
where *BaP_eq_* is the equivalent level of BaP; *TEF_i_* is the toxic equivalent factor of the *i*th PAH, and *C_i_* is the concentration of the *i*th PAH. Relevant information on PAHs is depicted in Table 1.

## 3. Results and Discussions

### 3.1. Characteristics of Personal PAHs Exposure

Polycyclic aromatic hydrocarbons concentrations based on personal PM_2.5_ samples were shown in Table 2. The total concentration of the detected 14 PAHs ranged from 34.0 to 472.7 ng/m^3^, with an average value of 106.4 ng/m^3^. The average exposure level of BaP was 8.0 ng/m^3^ and the calculated *BaP_eq_* was 15.3 ng/m^3^, accounting for 7.6% and 14.4% of the total PAHs, respectively. No significant differences in total PM_2.5_-bound PAH concentrations were found between ETS- and non-ETS (including passive smoking)-exposed people or cooking and non-cooking people (*p* > 0.05). Compared with two other studies conducted during winter in Tianjin, the levels of personal PM_2.5_-bound PAH exposure in the present study were nearly twice as high as those of children (∑PAHs 58.2 ng/m^3^, BaP 5.65 ng/m^3^) [14], but comparable with those of the PM_10_-bound PAH exposure of senior citizens (∑PAHs 162.4 ng/m^3^, BaP 9.0 ng/m^3^) [10]. Mu et al. [20] reported that the levels of personal PM_2.5_-bound PAH exposure for adults in Wuhan and Zhuhai during spring were 11.9 and 8.3 ng/m^3^, respectively, which were much lower than that of this study, suggesting the seasonal variation of PAH exposure. Compared with occupational exposure, the level of ∑PAH exposure of senior citizens was higher than that of vehicle inspection workers at gasoline lines (56.1 ng/m^3^), comparable with that of workers at bus lines (111.7 ng/m^3^), and lower than that of cooks in Chinese stall (141.0 ng/m^3^) and workers at diesel lines (199.8 ng/m^3^); but the levels of PM_2.5_-bound BaP and *BaP_eq_* of senior citizens were higher than those of the cooks and vehicle inspection workers [41,42]. The result was unexpected as vehicle emission could significantly increase the level of particulate PAH exposure [43,44]. The occupational exposure of cooks and vehicle inspection workers was represented by the concentration at the workplace. Therefore, the comparison suggested the existence of other sources emitting higher *BaP_eq_* per unit source contributions than cooking and vehicle emissions.

The HMW-PAHs were the primary PAH category (occupying 67.3% of total PAHs), and were followed by MMW-PAHs (27.0% of total PAHs) and LMW-PAHs (6.0% of total PAHs). The average ratio of LMW-PAHs/HMW-PAHs was 0.1. The three dominant PAH individuals were BbF (22.7 ± 14.9 ng/m^3^), IND (14.7 ± 8.8 ng/m^3^), and BghiP (11.7 ± 6.9 ng/m^3^), accounting for 21.4%, 14.3%, and 11.4% of total PAHs, respectively. The high proportion of HMW-PAHs and low ratio of LMW-PAHs/HMW-PAHs indicated emission from pyrogenic processes, such as coal combustion and vehicle emission [9]. The dominance of BbF, IND, and BghiP in this study was the same as that of personal PM_10_-bound PAH exposure for senior citizens [10] but different from that of the ambient atmosphere [45] during winter in Tianjin. See et al. [42] reported that stir-frying in Chinese stalls would give rise to an abundance of PM_2.5_-bound BbF, IND, and BghiP. Therefore, the influence of indoor PAHs sources might be of importance and exposure bias may also be reflected in the proportion of different PAH species.

### 3.2. Source Apportionment of Personal PAHs Exposure

EPA PMF5.0 was run in a random seed model with different numbers of factors. The corresponding *Q*/*Q_exp_* was obtained and the decrease from five-factor to six-factor was the largest. Therefore, six sources were determined in terms of senior citizens’ exposure to PM_2.5_-bound PAHs in winter and the source profile is depicted in Figure 1.

Factor 1 was dominated by DahA (63.8%) and moderately loaded by BaA (30.6%) and Chr (26.6%). DahA was used as the indicatory PAH for biomass burning [28], and BaA and Chr were good tracers for biomass burning [46,47]. Biomass was burned in an open field as agricultural waste or in stoves as fuel [29], which was not forbidden in Tianjin in 2011. Moreover, regional transport from nearby provinces, such as Hebei and Shandong, was considered [27]. Zhang et al. [29] reported that a 20% increase in the contribution of biomass burning in the Beijing–Tianjin area was from nearby provinces. Therefore, factor 1 was identified as biomass burning, contributing 14.7% to the PAHs’ mass concentration.

Factor 2 depicted the major loading of Phe (77.5%) and Fl (43.6%). Yao et al. [48] concluded that 3–4 ring PAHs were related to cooking emission. An abundance of Phe was found in commercial and domestic kitchen exhausts. Furthermore, it indicated liquid petroleum gas burning and cooking oil fumes together with Fl [49], as the relatively low temperature during cooking mainly produced lighter PAHs. Zhang et al. [37] established an emission profile of domestic cooking in China, suggesting that Phe and Fl could be recognized as source signatures of cooking. Moreover, the ratio of IND/(IND + BghiP) in this factor (0.7) was close to the reported value of 0.5 [37], displaying the impact of cooking emission. According to the time–activity survey, 71.3% of the elderly spent an average of 1.5 h cooking daily. Thus, Factor 2 was identified as cooking, which contributed 9.9% to the PAHs’ mass concentration.

Factor 3 was enriched with Pyr (58.3%), Chr (38.1%), and Fl (36.5%). Pyr was regarded as a source tracer of coal combustion [46,50]. Several studies revealed that Pyr, Chr, and Fl were dominant PAH species in coal combustion [5,17,27,46,51]. In 2011, 23.66 million tons of coal were consumed, accounting for over 50% of fossil fuel usage in Tianjin. Coal emissions from heating sources increased ambient PAH concentrations in winter [6,7,10,14]. Hence, factor 3 could represent coal combustion, which contributed 27.1% to the PAHs’ mass concentration.

For factor 4, there was a predominance of BkF (61.4%), and moderate Flu (38.9%), and BaP (37.4%). BkF showed significant dominance in the diesel emission profile and was often used as a diesel marker [15,46,49,50,52]. High fractions of Flu and BaP in diesel emission profiles were reported [15,17,50]. The ratio of BbF/BkF in the factor (1.5) was higher than 0.5, which could be used to indicate diesel emission [51]. Two arterial roads surrounded the selected communities, with a bus stop nearby. The senior citizens preferred buses for commuting, and diesel was the main fuel for buses in Tianjin in 2011. Hence, factor 4 was classified as diesel emission, which contributed 18.7% of the PAHs’ mass concentration.

Factor 5 was mainly loaded with BaP (27.0%), BaA (21.6%), and Fl (19.9%). Lu and Zhu [53] found that over 80% of residential Bap came from tobacco smoking. BaA was characterized as a source marker of environmental tobacco smoking [25,38]. Mannino and Orecchio [54] pointed out that the level of Fl was higher in smoking-permitting families than that in smoking-forbidding families. The ratio of BaA/BaA + Chr in the factor was 0.6, equal to that reported by Zhang et al. [38], which could be used to identify the ETS. Among the participants, 14.9% of the elderly were smokers, while 28.7% were exposed to ETS. Shang et al. [34] found that ETS sources could affect people with no cigarette exposure, suggesting that cigarette smoke could be present in ambient PM_2.5_. Therefore, factor 5 was regarded as ETS, which contributed 9.2% of the PAHs’ mass concentration.

Factor 6 was highly loaded with Ace (49.4%), Flu (49.0%), IND (39.4%), and BghiP (39.1%). The LWM-PAHs, such as Ace, were linked with unburned petroleum and gasoline emissions [50]. High loading of Flu indicated the emission of gasoline-powered vehicles [50,55]. In addition, IND and BghiP were typical PAHs in gasoline vehicle exhausts [17,49,56]. The ratio of Pyr/BaP in the factor was 0.6 [57], suggesting the presence of gasoline emission. Tianjin was experiencing a 20.6% car ownership growth rate in 2011; thus, factor 6 was considered as gasoline emission, which contributed 20.4% of the PAHs’ mass concentration.

Coal combustion, vehicle (gasoline + diesel) emission, and biomass burning contributed the most to personal PM_2.5_-bound PAH exposure, as reported previously [10,14], as they were primary ambient PAHs sources in the Beijing–Tianjin district [29]. Although there were no significant differences in total PM_2.5_-bound PAH concentration between ETS- and non-ETS (including passive smoking)-exposed people or cooking and non-cooking people (*p* > 0.05), respectively, the chemical compositions of exposed PAHs were influenced by these two sources. It was noted that indoor sources contributed nearly 20% to the senior citizen PAH exposure, which suggested that individual domestic activities should be considered in personal exposure estimation instead of the substitute of ambient monitoring data. In addition, unlike ambient sources, it was convenient and effective for the public to take actions, such as using range hoods during cooking [37] or preventing indoor smoking, to reduce the influence of indoor sources.

### 3.3. Carcinogenic Risk of PAH Sources via Inhalation

In this study, the carcinogenic risk induced by PM_2.5_-bound PAHs via inhalation was estimated for each contributing source based on personal exposure data. According to the US EPA, the acceptable limit of carcinogenic risk was 1.0 × 10^−6^, and the tolerance limit was 1.0 × 10^−4^ [39]. For senior citizens, the average inhalational carcinogenic risk of PAH exposure was 1.6 × 10^−5^, which was within the tolerable range for the public (1.0 × 10^−6^–1.0 × 10^−4^). As shown in Figure 2, the individual risk of the identified sources occurred in the following sequence: biomass burning > diesel emission > gasoline emission > ETS > coal combustion > cooking, with the risk value of 4.1 × 10^−6^, 3.8 × 10^−6^, 3.1 × 10^−6^, 3.0 × 10^−6^, 1.5 × 10^−6^, and 0.9 × 10^−6^, respectively. The contributions of biomass burning, diesel emission, ETS, and indoor sources (sum of cooking and ETS) to PAH-induced carcinogenic risk were higher than those to the PAH mass concentration, as shown in Table 3.

Biomass burning contributed the most to the risk of PM_2.5_-bound PAH exposure, as the toxic DahA and BaA were prominent in the biomass burning profiles [46,47]. Exposure to PAHs from biomass burning would increase the risk of lung cancer [58]. Andersen et al. [59] reported that PM_10_ emitted from biomass burning would induce higher daily respiratory hospital admissions than that emitted from traffic. The primary contribution to carcinogenic risk from diesel and gasoline emission was reported by Han et al. [10]. Hime et al. [36] concluded that vehicle emission might cause greater harm than other sources by reviewing the health effect of ambient PM emissions. In particular, diesel engine exhaust was classified as carcinogenic to humans [60]. ETS was the only individual source with the fifth percentile risk value higher than 1.0 × 10^−6^; ETS has been reported as the primary source of indoor BaP [38,53], which was found to be related to over 0.7 million deaths in China annually [61]. With approximately 300 million smokers in China, 740 million non-smokers are exposed to ETS [34]. Hence, the result of this study can raise public concern on the hazards of ETS exposure and promote the implementation of national cigarette control regulations. The proportion of coal combustion contributed less than 10% to exposure risk, which might be associated with a lower fraction of toxic PAH species in the coal combustion profile in Tianjin, as reported by Shi et al. [30]. The carcinogenic risk associated with cooking observed in this study (0.9 × 10^−6^) was comparable with that observed in households using the range hood for cooking (0.6 × 10^−6^) [37]. However, the 50th percentile risk value was over the acceptable risk limit, which should not be neglected. The contribution of outdoor sources to exposure risk was higher than that of indoor sources (76.1% vs. 23.9%), but the fifth percentile risk value of indoor and outdoor sources was over 1.0 × 10^−6^. Thus, the relatively higher carcinogenic risk per unit contribution of indoor sources was consistent with the suggestion that indoor air pollution had equal or even higher health risks compared with outdoor air pollution [33].

Due to the limited availability of resources, only senior citizens were recruited in this study, which would influence the generalization of the findings. The underestimation of health risk of PAHs in this study was unavoidable as only fourteen PAH species were analyzed, which could not represent the actual PAH exposure. In addition, the lack of vapor phase PAHs data might increase the uncertainty of PMF. Without considering the vapor and particle partitioning, the model would have a biased source profile. Besides, the health risk value was calculated based on the assumption of PAH exposure for preliminary estimation, which did not have a direct link with adverse health effects.

## 4. Conclusions

In this study, 101 senior citizens volunteered to participate in personal PM_2.5_ sample collection. In addition, the characteristics, source contributions, and carcinogenic risks of personal PM_2.5_-bound PAH exposure during winter in Tianjin were discussed. The results showed that the average exposure level of PM_2.5_-bound PAHs for the elderly was 106.4 ng/m^3^ in winter, and PAHs were mainly distributed in HMW range with BbF, IND, and BghiP being dominant. Six main emission sources were apportioned, including coal combustion (27.1%), gasoline emission (20.4%), diesel emission (18.7%), biomass burning (14.7%), cooking (9.9%), and ETS (9.2%). The average inhalational carcinogenic risk of PAH exposure was 1.6 × 10^−5^ in total, with indoor sources of 0.4 × 10^−5^ and outdoor sources of 1.2 × 10^−5^. The carcinogenic risk contribution of each source was in the following sequence: biomass burning (25.2%) > diesel emission (23.2%) > gasoline emission (18.7%) > ETS (18.5%) > coal combustion (9.0%) > cooking (5.4%). Moreover, the contributions of biomass burning, diesel emission, ETS, and indoor sources (cooking and ETS) to PAH-induced carcinogenic risk were higher than those to the PAH mass concentration, suggesting severe carcinogenic risk per unit contribution. This study can help regulators inform functional and cost-effective emission reduction policies, leasing to the reducing of personal PM_2.5_-bound PAH exposure, which could mitigate lung cancer risk.

## Figures and Tables

**Figure 1 ijerph-19-04440-f001:**
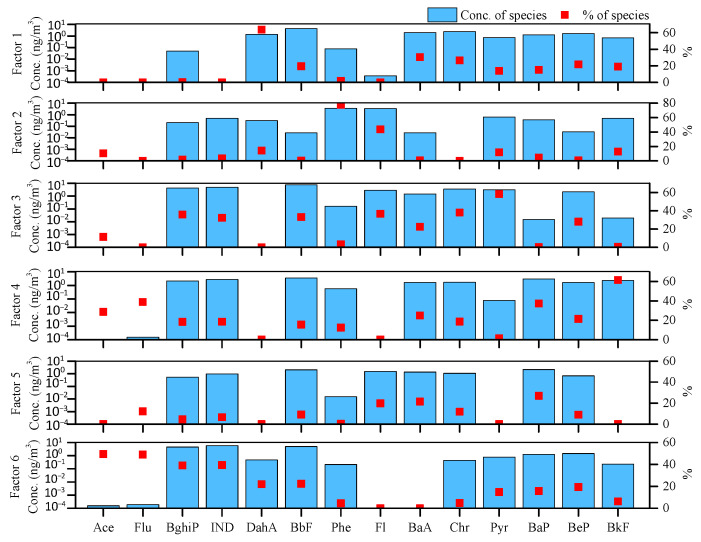
Source profile of personal PM_2.5_−bound PAH exposure. Conc., the abbreviation for concentration; % of species, the percentage of species; Ace, acenaphthene; Flu, fluorene, Phe, phenanthrene; Fl, fluoranthene; Pyr, pyrene; BaA, benz[a]anthracene; Chr, chrysene; BbF, benzo[b]fluoranthene; BkF, benzo[k]fluoranthene; BeP, benz[e]pyrene; BaP, benzo[a]pyrene; DahA, dibenz[a,h]anthracene; BghiP, benzo[ghi]perylene; IND, indeno[1,2,3-cd]pyrene.

**Figure 2 ijerph-19-04440-f002:**
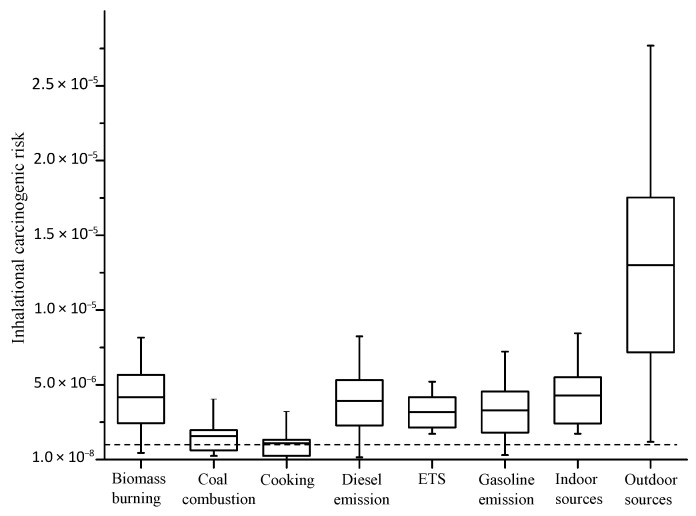
Box (25–75th percentiles) and whisker (5–95th percentiles) plot of inhalational carcinogenic risk of each source (horizontal lines in the middle of the boxes, median values; ETS, environmental tobacco smoking; dotted line, 1.0 × 10^−6^).

**Table 1 ijerph-19-04440-t001:** Properties of polycyclic aromatic hydrocarbons.

PAH Species	Abbreviation	Rings	MW Groups	TEFs ^a^	*IUR* ^b^ (μg/m^3^)^−1^
Acenaphthene	Ace	3	LMW	0.001	/
Fluorene	Flu	3	LMW	0.001	/
Phenanthrene	Phe	3	LMW	0.001	/
Fluoranthene	Fl	4	MMW	0.001	/
Pyrene	Pyr	4	MMW	0.001	/
Benz[a]anthracene	BaA	4	MMW	0.1	/
Chrysene	Chr	4	MMW	0.01	/
Benzo[b]fluoranthene	BbF	5	HMW	0.1	/
Benzo[k]fluoranthene	BkF	5	HMW	0.1	/
Benz[e]pyrene	BeP	5	HMW	/	/
Benzo[a]pyrene	BaP	5	HMW	1	1.1 × 10^−3^
Dibenz[a,h]anthracene	DahA	5	HMW	1	/
Benzo[ghi]perylene	BghiP	6	HMW	0.01	/
Indeno[1,2,3-cd]pyrene	IND	6	HMW	0.1	/

^a^ Nisbet and LaGoy, 1992; ^b^ EPA, 2011; MW, molecular weight; LMW, low molecular weight; MMW, middle molecular weight; HMW, high molecular weight.

**Table 2 ijerph-19-04440-t002:** Polycyclic aromatic hydrocarbons concentrations based on personal PM_2.5_ samples (ng/m^3^).

PAHs	All Population (*n* = 87)	ETS-Exposed (*n* = 27)	Non-ETS (*n* = 60)	Cooking (*n* = 52)	Non-Cooking (*n* = 35)
Ace	0.5 ± 1.2	0.7 ± 1.5	0.5 ± 1.1	0.5 ± 1.4	0.6 ± 1.1
Flu	0.7 ± 1.2	0.9 ± 1.5	0.7 ± 1.0	0.7 ± 1.2	0.8 ± 1.2
Phe	5.0 ± 4.5	5.4 ± 5.6	4.8 ± 3.9	4.8 ± 4.3	5.2 ± 4.7
Fl	8.1 ± 7.3	8.6 ± 8.9	7.9 ± 6.5	7.7 ± 6.5	8.7 ± 8.4
Pyr	5.7 ± 4.7	6.1 ± 6.1	5.6 ± 3.9	5.6 ± 4.4	6.0 ± 5.0
BaA	6.7 ± 5.3	7.2 ± 6.3	6.5 ± 4.8	6.4 ± 5.1	7.1 ± 5.6
Chr	9.2 ± 7.0	9.9 ± 9.4	8.9 ± 5.7	8.7 ± 6.2	9.9 ± 8.1
BghiP	11.7 ± 6.9	12.7 ± 8.7	11.2 ± 5.9	11.3 ± 6.7	12.3 ± 7.3
IND	14.7 ± 8.8	16.0 ± 11.4	14.1 ± 7.4	14.1 ± 8.4	15.7 ± 9.6
DahA	2.3 ± 1.8	2.5 ± 2.1	2.1 ± 1.7	2.1 ± 1.8	2.5 ± 1.8
BbF	22.7 ± 14.9	24.1 ± 19.1	22.1 ± 12.6	21.8 ± 13.7	24.0 ± 16.6
BaP	8.0 ± 5.4	8.3 ± 5.9	7.9 ± 5.2	7.7 ± 5.5	8.5 ± 5.3
BeP	7.5 ± 4.7	7.7 ± 6.1	7.4 ± 4.0	7.2 ± 4.3	8.0 ± 5.3
BkF	3.7 ± 2.5	3.6 ± 3.1	3.7 ± 2.3	3.6 ± 2.4	3.9 ± 2.7
∑PAHs	106.4 ± 70.9	113.4 ± 91.0	103.3 ± 60.4	102.1 ± 66.5	112.9 ± 77.6
*BaP_eq_*	15.3 ± 10.1	16.2 ± 12.1	14.9 ± 9.2	14.6 ± 9.9	16.3 ± 10.5

ETS, environmental tobacco smoking; SD, standard deviation; Ace, acenaphthene; Flu, fluorene; Phe, phenanthrene; Fl, fluoranthene; Pyr, pyrene; BaA, benz[a]anthracene; Chr, chrysene; BbF, benzo[b]fluoranthene; BkF, benzo[k]fluoranthene; BeP, benz[e]pyrene; BaP, benzo[a]pyrene; DahA, dibenz[a,h]anthracene; BghiP, benzo[ghi]perylene; IND, indeno[1,2,3-cd]pyrene.

**Table 3 ijerph-19-04440-t003:** Comparison of contribution to mass concentration and carcinogenic risk of each source.

	Contribution to Carcinogenic Risk (%)	Contribution to Mass Concentration (%)	Ratio of Contribution to Carcinogenic Risk/ Contribution to Mass Concentration
Biomass burning	25.2	14.7	1.7
Coal combustion	9.0	27.1	0.3
Cooking	5.4	9.9	0.5
Diesel emission	23.2	18.7	1.2
ETS	18.5	9.2	2.0
Gasoline emission	18.7	20.4	0.9
Indoor	23.9	19.1	1.2
Outdoor	76.1	80.9	0.9

ETS, environmental tobacco smoking.

## Data Availability

The data presented in this study are available on request from the corresponding author.

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
