# Peer review of "Characteristics, Source Contributions, and Source-Specific Health Risks of PM2.5-Bound Polycyclic Aromatic Hydrocarbons for Senior Citizens during the Heating Season in Tianjin, China"

_ijerph, 2022, doi:10.3390/ijerph19084440_

Round 1

Reviewer 1 Report

The paper presents the study of elderly personal exposure to PM2.5 and associated carcinogenic risks caused by PM2.5-bound PAHs. The dedicated experimental campaign was conducted in Tianjin, China during the winter period in 2011. The total of 101 senior citizens were recruited for personal PM2.5 sample collection (November 30 to December 12) and fourteen PAHs were analyzed. The standard receptor model Positive Matrix Factorization (PMF) was used for source apportionment and six main emission sources were identified (coal combustion, gasoline emission, diesel emission, biomass burning, cooking and environmental tobacco smoking). In addition, inhalation carcinogenic risks of each source were estimated.

Generally, the paper refers to the very important air quality problem and from my point of view the presented results are very interesting and could be of wider interest. Although there is no new scientific approach, and standard analysis have been used, the major contribution of the presented results is related to the assessment of the specific personal exposure and can be a good starting point for further similar extended studies.

Please find below several suggestions that can be used for improving the manuscript

Lines 105-106:  "During volunteer recruitment, questionnaires were distributed to the -senior citizens, containing information such as name, sex, age, living habits, and health condition"

I suggest to add more information and analysis using these information (is it only used for selection of volunteers?)

How do living habits contribute to the PM samples – is there any assessment. What is the difference in sampling if senior citizens use personal monitor samples with respect to other citizens? Is there any bias? The authors should briefly comment on these issues.

Table S1: Time of activity (h) – is it per day? Please add information and clarify it. If the sample is associated with 24h it is strange to declare 24h indoor activity (max). If so, the authors should comment the contribution of outdoor sources – how reliable are the samples i.e. did the authors use any additional information in the analysis ? (time of the activity is presented but it is not used in the discussion – is there any significant influences on the PM and PAHs results?.

In the Sample collection paragraph this issue should be also better explained – is one volunteer associated with only one PM sample or more? How does it influence the representatives of the samples?

PMF – I suggest to add information if the rotation of the factors was used, what is the alpha value?

Lines 194-196: “This section may be divided by subheadings. It should provide a concise and precise description of the experimental results, their interpretation, as well as the experimental conclusions that can be drawn.”

This paragraph is probably from the journal template and should be removed.

Typing errors:

Lines 25, 26, 36, 37: Please use subscript for 2.5 in PM2.5, to be consistent through the manuscript

Please use the uniform style for units through the manuscript i.e.,   ng/m3 or ngm-3

Figure 2: It is hard to read the text, font should be increased

Author Response

Dear Editor and reviewer,

We received the comments back from the reviewers for our paper entitled “Characteristics, source contributions, and source-specific health risks of PM2.5-bound polycyclic aromatic hydrocarbons for senior citizens during the heating season in Tianjin, China” (Manuscript number: ijerph-1644907). We have carefully considered all of the suggestions and comments made by the reviewers, which were all pertinent and useful, and have substantially revised our manuscript. Attached you can find a point by point response to all comments along with a description of the changes incorporated in the manuscript. The changes are marked in red in the manuscript.

If we have not addressed these issues properly, please feel free to contact us. We will alter the paper as needed to get this work published in International Journal of Environmental Research and Public Health. Please contact us if you have any further questions.

Reviewer 2 Report

Interesting paper with useful results.  I have several minor comments.

There is no mention of meteorology.  Tianjin is not terribly far from non-local pollution sources such as Beijing, and from meteorological influences like sea breezes from the Yellow Sea.  Is it possible that the sensors were sometimes impacted by particles from these more distant sources?  I'd like to see at least a brief discussion or speculation about this.

Line 101 "There were no evident stationary pollution sources nearby." - this seems unrealistic.  No gas stations with fugitive VOC emissions?  No stationary emissions due to restaurants or food vendors?

Line 240: please briefly explain the "random seed model".

Figure 1: Needs to be improved for clarity.  "Con" should be defined.  What is meant by "%"?  (I.e., percentage of what?)  Also, the concentration axis spans an incredible five orders of magnitude!  Is this common?  Or unusual?  It seems worthy of discussion at least.

Figure 2: is the horizontal line in the middle of the boxes the median?  Or the mean?  Please clarify this in the figure caption.  Also, I have the same comment about the logarithmic scale as for Figure 1.  With this scaling, the median values of the Biomass burning and ETS distributions may differ by nearly a factor of 2, but that difference is obscured if a linear scale is not used.

Author Response

(The authors gave the same response as above.)

Round 2

Reviewer 1 Report

The authors revised the manuscript following most of the previous suggestions and reasonably explained several issues.
From my point of view the presented results could be interesting for wider community and it deserves publishing